# RaCNN: Region-aware Convolutional Neural Network with Global Receptive Field

## Abstract

Recent Convolutional Neural Networks (CNNs) utilize large-kernel convolutions (e.g., 101 kernel convolutions) to simulate a large receptive field of Vision Transformers (ViTs). However, these models introduce specialized techniques like reparameterization, sparsity, and weight decomposition, increasing the complexity of the training and inference stages. To address this challenge, we propose Region-aware CNN (RaCNN), which achieves a global receptive field without requiring extra complexity, yet surpasses state-of-the-art models. Specifically, we design two novel modules to capture global visual dependencies. The first is the Region-aware Feed Forward Network (RaFFN). It uses a novel Region Point-Wise Convolution (RPWConv) to capture global visual cues in a region-aware manner. In contrast, traditional PWConv shares the same weights for all spatial pixels and cannot capture spatial information. The second is the Region-aware Gated Linear Unit (RaGLU). This channel mixer captures long-range visual dependencies in a sparse global manner and can become a better substitute for the original FFN. Under only 84% computational complexity, RaCNN significantly outperforms the state-of-the-art CNN model MogaNet (83.9% vs. 83.4%). It also demonstrates good scalability and surpasses existing state-of-the-art lightweight models. Furthermore, our RaCNN shows comparability with state-of-the-art ViTs, MLPs, and Mambas in object detection, instance segmentation, and semantic segmentation. All codes and logs are released in the supplementary materials.

## 1 Introduction

Convolutional Neural Networks (CNNs) have been one of the most important fields in computer vision over the past decade. Pioneering works like AlexNet (Krizhevsky et al., 2012) use large kernels to improve performance. After that, ResNet (He et al., 2016) applies small kernels and achieves leading performance through residual connections, establishing the dominant position of small-kernel CNNs in the vision domain. Recently, Vision Transformers (ViTs) Dosovitskiy et al. (2021); Liu et al. (2021); Hassani et al. (2023a) have obtained great success in vision by capturing the global receptive field. Inspired by this, recent CNNs have utilized large-kernel convolutions (e.g., 101 kernels (Chen et al., 2024)) to simulate the large receptive fields of Vision Transformers (ViTs). Both ConvNeXt (Liu et al.,

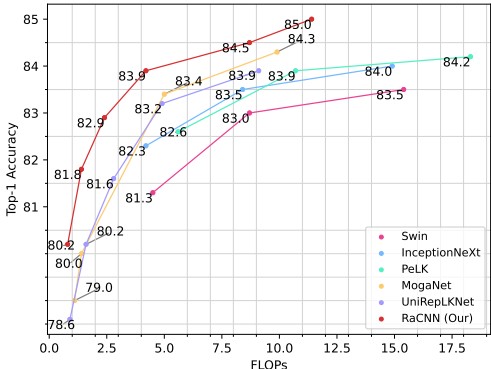

Figure 1: Comparing the accuracy and FLOPs with Swin (Liu et al., 2021), InceptionNeXt (Yu et al., 2024), PeLK (Chen et al., 2024), MogaNet (Li et al., 2024), and UniRepLKNet (Ding et al., 2024) on ImageNet-1K.

2022) and DWNet (Han et al., 2022) find using $7 \times 7$ kernels can obtain even better results than Swin (Liu et al., 2021). RepLKNet (Ding et al., 2022) proposes replacing commonly used small kernels with large Depth-Wise Convolution (DWConv) up to $31 \times 31$ to obtain a larger receptive field, followed by more works attempting to increase the kernel size further (e.g., 51 in SLaK Liu et al. (2023), and 101 in PeLK (Chen et al., 2024)). Large-kernel convolutions endow CNNs with powerful capabilities, achieving comparable or superior accuracy to ViTs while maintaining higher

| Reference | Method | Max KS | Throughput (img/s) | Param | FLOPs | Top-1 (%) |
|---|---|---|---|---|---|---|
| CVPR22 | ConvNeXt (Liu et al., 2022) | 7 | 2729 | 29M | 4.5G | 82.1 |
| NeurIPS22 | FocalNet (LRF) (Yang et al., 2022) | 7 | 2443 | 29M | 4.5G | 82.3 |
| ICLR23 | ConvNeXt-dcls (Hassani et al., 2023b) | 17 | 1585 | 29M | 5.0G | 82.5 |
| ICLR23 | SLaK (Liu et al., 2023) | 51 | 417 | 30M | 5.0G | 82.5 |
| ICCV23 | ConvNeXt-1D++ (Kirchmeyer & Deng, 2023) | 31 | 1043 | 29M | 4.7G | 82.7 |
| CVM23 | VAN (Guo et al., 2023) | 21 | 1688 | 27M | 5.0G | 82.8 |
| **Our** | **RaCNN-T** | Global | **3037** | 19M | 2.4G | **82.9** |
| CVPR23 | ConvNeXt V2 (Woo et al., 2023) | 7 | 1396 | 29M | 4.5G | 83.0 |
| NeurIPS22 | HorNet (Rao et al., 2022) | 7 | 1417 | 22M | 4.0G | 82.8 |
| ICLR22 | DWNet (Han et al., 2022) | 7 | 1741 | 74M | 12.9G | 83.2 |
| CVPR24 | UniRepLKNet (Ding et al., 2024) | 7 | 1101 | 31M | 4.9G | 83.2 |
| ICLR24 | MogaNet (Li et al., 2024) | 7 | 1171 | 25M | 5.0G | 83.4 |
| CVPR22 | RepLKNet (Ding et al., 2022) | 31 | 585 | 79M | 15.3G | 83.5 |
| CVPR24 | InceptionNeXt (Yu et al., 2024) | 11 | 2164 | 49M | 8.4G | 83.5 |
| **Our** | **RaCNN-S** | Global | **2185** | 28M | 4.2G | **83.9** |

Table 1: Comparison of various CNNs on ImageNet-1K image classification. Throughput is tested on a 4090 GPU with 128 batch size and BN merge. **Max KS** is the abbreviation of the Max Kernel Size of convolution. We show two scales of RaCNN to compare with others. Most state-of-the-art CNNs introduce large-kernel convolutions to obtain better results, and our RaCNN obtains the best results with sparse global kernel size.

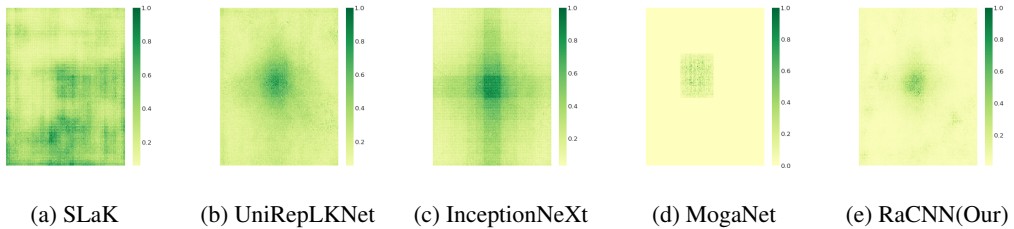

| (a) SLaK | (b) UniRepLKNet | (c) InceptionNeXt | (d) MogaNet | (e) RaCNN(Our) |
|---|---|---|---|---|

Figure 2: **Effective receptive field (ERF) of various CNNs.** SLak (Liu et al., 2023), UnirepLKNet Ding et al. (2024), and InceptionNeXt (Yu et al., 2024) could capture long-range dependencies but introduce excessive visual noises, as they allocate excessive weights to background and edge areas. MogaNet (Li et al., 2024) could only capture local visual cues. Our RaCNN can capture long-range dependencies and the local context features simultaneously without excessive noises.

efficiency. This renaissance emphasizes the potential of CNNs and highlights the importance of a large receptive field in vision perception.

However, the quadratic complexity of the kernel size seriously hinders the efficiency of large-kernel CNNs. As shown in Table 1, increasing the kernel size will add computational complexity, making it difficult to train such models. RepLKNet (Ding et al., 2022) and UniRepLKNet (Ding et al., 2024) propose re-parameterizing small kernels into larger ones. SLaK (Liu et al., 2023) uses two stripe convolutions and leverages dynamic sparsity to obtain trainable large kernels. Inception-NeXt (Yu et al., 2024) decomposes weights into several branches of smaller kernels in the Inception style (Szegedy et al., 2015). All the above compensatory measures introduce additional complexity during training and inference. As a result, these works remain conservative and cautious when expanding the receptive field, impeding further exploration and experimentation.

The above discussion leads to the following question: **Can we scale up the receptive field as much as possible without extra complexity?** To address this challenge, we present Region-aware CNN (RaCNN), a large-kernel CNN that provides a global receptive field without specialized training techniques. Specifically, we introduce two innovative modules to capture long-range dependencies. We first design the Region-aware Feed Forward Network (RaFFN) with novel Region Point-Wise Convolution (RPWConv) to capture global visual cues in a region-aware manner. Traditional PW-Conv Howard et al. (2017) is essentially a $1 \times 1$ convolution where all spatial pixels share the same weights, diminishing its ability to aggregate spatial information. In contrast, our RPWConv divides spatial feature maps into several sparse global regions, and generates dynamic weights within each region, exhibiting a coarse-grained global spatial recognition capability. The second proposed module is the Region-aware Gated Linear Unit (RaGLU), which captures long-range visual dependencies at a lower feature resolution, and can effectively replace the original Feed-Forward Network. The

above modules expand the receptive field of convolutions to a global scale, as illustrated in Figure 2. Unlike other CNNs, which either lose global visual cues or are susceptible to visual noise, our RaCNN simultaneously captures global cues and local features with high robustness and low complexity.

Our RaCNN impressively achieves leading performance compared with various architectures across various visual tasks. RaCNN substantially surpasses the state-of-the-art CNN model Inception-NeXt (Yu et al., 2024) (83.9% vs 83.5%) on ImageNet-1K image classification, while using only half the FLOPs (4.2G vs 8.4G) and gaining a slightly faster training speed. Furthermore, RaCNN exhibits satisfying scalability, outperforming existing state-of-the-art lightweight models. When used as a vision backbone, RaCNN also demonstrates performance comparable to state-of-the-art ViTs, MLPs, and Mambas in object detection, instance segmentation, and semantic segmentation, highlighting its remarkable capability in dense prediction tasks.

## 2 RELATED WORK

### 2.1 LARGE-KERNEL CNNS

Large-kernel convolutions ($7 \times 7$ and $11 \times 11$) are commonly utilized in old-fashioned CNNs such as AlexNet (Krizhevsky et al., 2012) and Inception (Szegedy et al., 2015; Ioffe & Szegedy, 2015; Szegedy et al., 2016; 2017). VGG (Simonyan & Zisserman, 2015) proposes stacking several small kernels ($3 \times 3$ and $1 \times 1$) deeply to get large receptive fields and achieve better results. After that, large-kernel convolutions gradually fade away, and only some Neural Architecture Search-based models try to incorporate them, like MobileNetV3 (Howard et al., 2019) and EfficientNet (Tan & Le, 2019). Recently, inspired by the popularity of ViT (Dosovitskiy et al., 2021; Liu et al., 2021) in modeling long-range dependencies, ConvNeXt (Liu et al., 2022) follows the design paradigm of the Swin Transformer by modernizing a standard ResNet to a ViT, and it applies large $7 \times 7$ DW-Conv to achieve competitive performance. HorNet (Rao et al., 2022), DWNet (Han et al., 2022), and UniRepLKNet (Ding et al., 2024) also verify the validity of $7 \times 7$ kernel size in various tasks. Subsequently, many works further increase the kernel sizes. RepLKNet (Ding et al., 2022) enlarges the kernel to $31 \times 31$ and proposes a re-parameterization technique to solve training issues of large kernels. SLaK (Liu et al., 2023) combines two strip convolutions ($51 \times 5$ and $5 \times 51$) with dynamic sparsity to scale kernels up to $51 \times 51$. PeLK (Chen et al., 2024) pushes this further by incorporating parameter sharing to imitate human peripheral vision, which increases the kernel size to an astonishing $101 \times 101$. Our work further maximizes the kernels to the fullest to achieve the global receptive field free of specialized training tricks.

### 2.2 VISION TRANSFORMERS

Following the success of Transformer (Vaswani et al., 2017b) in NLP, Vision Transformer (ViT) (Dosovitskiy et al., 2021) demonstrates outstanding performance in image classification on ImageNet. Numerous follow-up works strive to enhance the performance of ViT. The well-known Swin Transformer (Liu et al., 2021) proposes shifted window attention, combining the attention mechanism with local windows, which remarkably boosts performance in various downstream vision tasks. Similarly, CSwin (Dong et al., 2022) computes self-attention in horizontal and vertical stripes in parallel to achieve better results with less computation. SMT (Lin et al., 2023) introduces multi-scale convolution to capture more local visual cues. Inspired by CNNs, NAT (Hassani et al., 2023a) proposes a variant of window-based attention to compute neighborhood attention in a sliding window manner, thus capturing sufficient local information for every spatial position. These novel variants of ViTs introduce inductive bias to improve results but only have a local receptive field within one block, instead of a global one in previous ViTs (Dosovitskiy et al., 2021; Touvron et al., 2021). Our RaCNN captures global visual dependencies in a block, thus allowing it to interact with long-range visual tokens.

### 2.3 VISION MLPS

Multilayer Perceptron (MLP) is a classical algorithm in the pre-CNN era. Recently, Channel MLP has become a core component in ViTs (Dosovitskiy et al., 2021; Touvron et al., 2021). Consequently,

some modern MLP-based architectures (Tolstikhin et al., 2021; Touvron et al., 2023) have been proposed to mix spatial features further and are even comparable to ViT (Dosovitskiy et al., 2021). To enhance the performance under limited computation, ViP (Hou et al., 2023), sMLPNet (Tang et al., 2022a), and Strip-MLP (Cao et al., 2023) decompose spatial mixing in two independent vertical and horizontal dimensions. However, all the aforementioned MLPs only process fixed-dimensional inputs and cannot generalize to downstream dense prediction tasks. Thus, researchers have replaced spatial MLPs with other spatial aggregation operations. AS-MLP (Lian et al., 2022), S2-MLP (Yu et al., 2022), Shift (Wang et al., 2022), and Hire-MLP (Guo et al., 2022) propose a spatial shift operation to aggregate spatial features. CycleMLP (Chen et al., 2023), Wave-MLP (Tang et al., 2022b), ATMNet (Wei et al., 2023), and RaMLP (Lai et al., 2023) use DWConv to introduce more fine-grained visual cues. Most of the above variants focus more on local information but lose the global context. Our RaCNN, in comparison, can capture global visual dependencies in one block.

## 2.4 VISION MAMBA

Mamba (Gu & Dao, 2023; Dao & Gu, 2024) is a recent advancement in sequence modeling that addresses the limitations of Transformer-based architectures and showcases new state-of-the-art performance. One of the key innovations of Mamba is the Selective State Space Model (SSM), which allows Mamba to manage long sequences more efficiently, scaling better with sequence length with lower complexity. Subsequent efforts (Huang et al., 2024; Pei et al., 2024; Liu et al., 2024; Shi et al., 2024; Yang et al., 2024; Zhu et al., 2024) have explored the adaptation of this block to vision tasks, yielding competitive results compared to other vision backbones. A direct approach is using different scanning routes to flatten 2D feature maps into 1D sequences, which are then modeled with the block and integrated. Inspired by these considerations, various scanning routes have been employed and proven to be effective, as evidenced by multiple studies. Our RaCNN models long-range dependencies in parallel, thus obtaining better training and inference speed.

## 3 METHOD

In this section, we first describe the overall architecture of RaCNN. Next, we show details of the Region-aware Feed Forward Network (RaFFN) and the Region-aware Gated Linear Unit (RaGLU). Finally, we describe several architecture variants of the RaCNN.

### 3.1 OVERALL ARCHITECTURE

Based on our proposed RaFFN and RaGLU, we build a series of architectures of different sizes, collectively dubbed Region-aware Convolution Neural Network (RaCNN). Figure 3 illustrates the architecture of RaCNN-Tiny. Following the ConvNeXt (Liu et al., 2022) framework, we construct a 4-stage architecture. The stem at the beginning is a convolutional layer with $3 \times 3$ kernels and a stride of 2, providing an effect of $2\times$ downsampling. Each stage contains a Down Block and several Region-aware (Ra) blocks. In all these blocks, PWConvs are commonly applied to facilitate inter-channel communications. Specifically, the Down Block reduces the input along the height and width dimensions, and increases the channel dimension using DWConv with $3 \times 3$ kernels of step 2 and a skip path. The Ra block consists of RaGLU and RaFFN, and these two modules do not change the feature size. RaGLU applies the Region Attention to mix different channels and capture the global context in a sparse global manner. In place of vanilla FFNs, RaFFN utilizes RPWConv and DWConv to further refine the global-aware features dynamically and carefully.

### 3.2 REGION-AWARE FEED FORWARD NETWORK

As shown in Figure 3, the RaFFN first feeds the input into a layer normalization to prevent numeric overflow issues. Then, the normalized features are fed into two parallel branches. One branch includes a PWConv and a Region PWConv (RPWConv), while the other consists of a PWConv and a DWConv. By adding the outputs of these two branches, we obtain multiscale features containing both local and global visual cues. The final output is generated simply by a residual connection, followed by GELU activation and another PWConv.

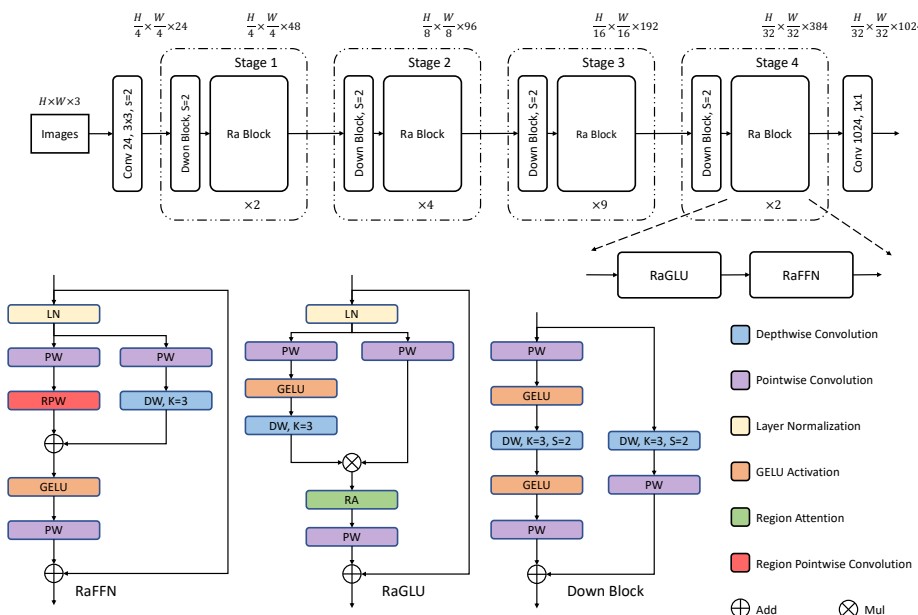

Figure 3: **Overview of RaCNN-Tiny.** It is constructed by stacking Region-aware (Ra) blocks. In each Ra block, the RaGLU module captures the global context, the RaFFN module aims to refine features dynamically and carefully.

Formally, consider a feature $x \in R^{c \times h \times w}$. The tensor flow in RaFFN can be elaborated as follows:

$$y^l = \text{LN}(x^{l-1}), \tag{1}$$

$$z_1^l = \text{RPW}(\text{PW}(y^l), rs), \tag{2}$$

$$z_2^l = \text{DW}(\text{PW}(y^l), ks = 3), \tag{3}$$

$$x^l = x^{l-1} + \text{PW}(\text{GELU}(z_1^l + z_2^l)), \tag{4}$$

where $l$ denotes the $l_{th}$ RaFFN. LN, PW, and GELU refer to Layer Normalization, PWConv, and GELU activation, respectively. $\text{RPW}(\cdot, rs)$ indicates the RPWConv with region size $rs$, and $\text{DW}(\cdot, ks = 3)$ represents the DWConv with kernel size 3. All DWConv and PWConv operations are followed by Batch Normalization (BN), which is not explicitly labeled for simplicity.

**Region Point-Wise Convolution:**  Figure 4a illustrates a traditional PWConv, which has been widely used in previous models (Simonyan & Zisserman, 2015; He et al., 2016) to exchange channel information. After finishing model training, the weights in PWConv become static. Thus, all inputs share the same weights in all spatial positions, leading to incompatibility with some hard cases. Dynamic PWConv, as shown in Figure 4b, is a variant of PWConv. The input generates its weight matrix; thus, it could be adaptively adjusted according to the input to capture visual dependencies better. Formally, consider a feature $x^l \in R^{c \times h \times w}$. The tensor flow in Dynamic PWConv can be elaborated as follows:

$$x = \text{Reshape}(x^l) \in R^{c \times hw}, \tag{5}$$

$$w = \text{Softmax}(s \frac{xx^T}{||x||_2^2}), \tag{6}$$

$$y = \text{Reshape}(wx) \in R^{c \times h \times w}, \tag{7}$$

where $l$ is the $l_{th}$ operation, $s$ is the learnable parameter to scale the similarity score, $w$ is the generated dynamic weight, and $y$ is the output.

However, Dynamic PWConv only generates weights for different inputs, but still shares the same weight for all positions within one input, which limits its ability to capture spatial information and expand the receptive field. To tackle this problem, we propose Region PWConv. Same as Swin

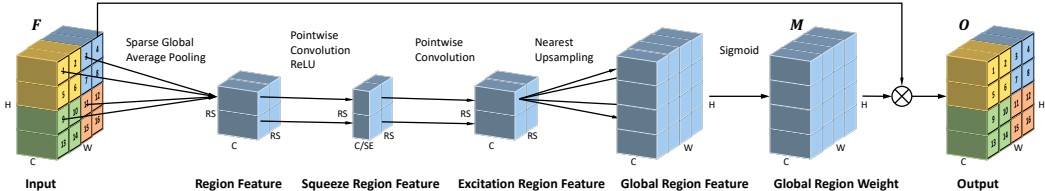

(a) PWConv        (b) Dynamic PWConv        (c) Region PWConv

Figure 4: **Comparison of various Point-Wise Convolutions.** (a) In PWConv, all inputs share the same static weight in all spatial positions. (b) Dynamic PWConv tailors weights for different inputs, but all spatial positions in a given input share the same weight. (c) Region PWConv partitions spatial features into several sparse global windows. Positions with the same color form one such window. Dynamic PWConv is applied in each window to generate region-aware dynamic weights.

Figure 5: **Detail of Region Attention.** Sparse global average pooling is applied in each sparse global window, aggregating global cues into each element in region feature. After PWConvs, upsampling and sigmoid, the obtained global region weight contains global cues and is later fused with the input.

Transformer (Liu et al., 2021), we partition the visual tokens into several regular windows and perform Dynamic PWConv within these windows respectively. Therefore, tokens in one input will get different weights. Figure 4c shows the details of the window partitioning. Instead of employing a local window approach like Swin, we adopt a dilated manner to capture global spare information and obtain a global receptive field while implementing Dynamic PWConv in each window.

**Comparison with Self-Attention:** The core of our model is generating dynamic weights, which is similar to self-attention (Vaswani et al., 2017a). Below, we outline the differences between them:

- Self-Attention requires three linear layers to project the input to different embeddings: Query, Key, and Value. Our model eliminates these layers and shares the same input.

- Self-Attention employs inner-product to calculate similarity, whereas our model used cosine distance to better measure the similarity.

- Self-Attention has quadratic computational complexity relative to the input image size, while the computational complexity of our model is linear to image size.

### 3.3 REGION-AWARE GATED LINEAR UNIT

Figure 3 shows the RaGLU, a two-branch residual architecture. The input is first processed through a Layer Normalization and then sent to two branches simultaneously. One branch consists of a PW-Conv, a GELU, and a DWConv. Another branch includes only a single PWConv. Then we multiply the outputs of two branches and pass the result through a Region Attention and a PWConv. Finally, we perform a residual connection between the output and the input to produce the final output. Mathematically, consider the input feature $\hat{x} \in R^{c \times h \times w}$. The whole process can be formulated as:

$$y^l = \text{LN}(x^{x-1}), \tag{8}$$

$$z_1^l = \text{DW}(\text{GELU}(\text{PW}(y^l)), ks = 3) \tag{9}$$

$$z_2^l = \text{PW}(y^l), \tag{10}$$

$$x^l = x^{l-1} + \text{PW}(\text{RA}(z_1^l \times z_2^l, rs)) \tag{11}$$

where $l$ is the $l_{th}$ GbR module. The notations LN, PW, and GELU mean Layer Normalization, PWConv, and GELU activation, respectively. $\text{DW}(\cdot, ks = 3)$ represents DWConv with kernel size

3, and RA($\cdot, rs$) indicates Region Attention with region size $rs$. All DWConv and PWConv are followed by BN operation, and we do not make explicit labeling for convenience.

**Region Attention (RA):** Squeeze-and-Excitation (SE) (Hu et al., 2018) is a famous channel attention, but it loses spatial prior due to compressing all spatial features into a single embedding. RA is a variant of the channel attention, and can better retain global visual cues because it maintains spatial prior by generating multiple embeddings.

As shown in Figure 5, the key difference between SE and RA is the use of sparse global average pooling. RA averages spatial features in a dilated manner, thereby generating several visual embeddings with spatial prior and a global receptive field.

### 3.4 MODEL VARIANTS

The architecture hyperparameters of these model variants are:

- RaCNN-P: C={24, 48, 96, 192}, L={2, 3, 8, 2}, R=8.0, Drop=0.00, Mix=0.1, Cut=0.2.
- RaCNN-N: C={32, 64, 128, 256}, L={3, 5, 8, 3}, R=6.0, Drop=0.05, Mix=0.2, Cut=0.3.
- RaCNN-T: C={48, 96, 192, 384}, L={3, 5, 10, 3}, R=4.0, Drop=0.10, Mix=0.4, Cut=0.5.
- RaCNN-S: C={64, 128, 256, 512}, L={3, 6, 14, 3}, R=3.0, Drop=0.20, Mix=0.8, Cut=1.0.
- RaCNN-B: C={96, 192, 384, 768}, L={4, 8, 16, 4}, R=2.0, Drop=0.35, Mix=0.8, Cut=1.0.

Here, $C$ is the embedding dimension of tokens, $L$ is the number of layers in Ra block, and $R$ is the expansion ratio for the RaGLU. Drop is the drop path rate during training, and Mix and Cut mean the Mixup and Cutmix ratio during training. Besides, the region size for the RPWConv and RaGLU $S$ is {8, 4, 2, 1}, and the head number $N$ for the RPWConv is set to {2, 4, 8, 16} for all variants.

### 4 EXPERIMENTS

#### 4.1 IMAGE CLASSIFICATION

**Settings.** We evaluate the RaCNN on ImageNet-1K (Deng et al., 2009) on 8 4090 GPUs. The training and augmentation strategies remain the same as ConvNeXt (Liu et al., 2022).

**Comparison with CNN-based Models.** The comparison of experimental results between RaCNN and other CNN-based models from recent years is presented in Table 2a. First of all, our breakthrough in image classification is noticeable. MogaNet-B (Li et al., 2024), the previous state-of-the-art CNN, uses three kernel sizes (3, 5, 7) to reach 84.3% accuracy with 9.9G FLOPs. In comparison, our RaCNN-B delivers 0.2% higher performance while requiring less than 85% of the computational cost. Moreover, compared with the large-kernel-based PeLK-B (Chen et al., 2024) that uses 51-size kernels, RaCNN-B achieves 0.3% higher accuracy with less than half computation (8.7G vs 18.3G), demonstrating its superiority and efficiency over previous large-kernel models.

**Comparison with SOTA Models.** Table 2b compares RaCNN with other state-of-the-art backbones, including Mamba-based, MLP-based and ViT-based models. When the model capacity is below 3G FLOPs, our RaCNN surpasses SiMBA (Patro & Agneeswaran, 2024), Wave-MLP (Tang et al., 2022b) and TransNeXt (Shi, 2024), all with similar FLOPs. For model capacities ranging from 4G to 11G FLOPs, RaCNN outperforms state-of-the-art Mamba-based models (VMamba (Liu et al., 2024) and SiMBA (Patro & Agneeswaran, 2024)), MLP-based models (RaMLP (Lai et al., 2023) and Wave-MLP (Tang et al., 2022b)) and ViT-based models (NAT (Hassani et al., 2023a) and BiFormer (Zhu et al., 2023)) with comparable FLOPs. For large models exceeding 11G FLOPs, RaCNN achieves higher performance than state-of-the-art architectures such as VMamba (Liu et al., 2024), RaMLP (Lai et al., 2023) and NAT (Hassani et al., 2023a).

**Comparison with Lightweight Models.** We further evaluate RaCNN-P against lightweight models, as shown in Table 3, and RaCNN delivers a significant performance margin. Compared to smaller lightweight models with less than 1G FLOPs, RaCNN has an advantage of 1.6%, significantly outperforming state-of-the-art models such as SwiftFormer (Shaker et al., 2023) and UniRepLKNet-F (Ding et al., 2024). For lightweight models with more than 1G FLOPs, RaCNN achieves at least

|  | (a) | | | | |  | (b) | | | |
|---|---|---|---|---|---|---|---|---|---|---|
| Models | Kernel | Top1 | FLOPs | Params | | Models | Arch. | Top1 | FLOPs | Params |
| DWNet | 7 | 81.3 | 3.8G | 24M | | SiMBA-S | Mamba | 81.7 | 2.4G | 15M |
| DWNet | 7 | 83.2 | 12.9G | 74M | | CycleMLP-B1 | MLP | 78.9 | 2.1G | 15M |
| ConvNeXt-T | 7 | 82.1 | 4.5G | 29M | | ATMNet-xT | MLP | 79.7 | 2.2G | 15M |
| ConvNeXt-S | 7 | 83.1 | 8.7G | 50M | | Wave-MLP-T | MLP | 80.6 | 2.4G | 17M |
| ConvNeXt-B | 7 | 83.8 | 15.4G | 89M | | BiFormer-T | ViT | 81.4 | 2.2G | 13M |
| HorNet-T | 7 | 82.8 | 4.0G | 22M | | NAT-M | ViT | 81.8 | 2.7G | 20M |
| HorNet-S | 7 | 83.8 | 8.8G | 50M | | SMT-T | ViT | 82.2 | 2.4G | 12M |
| HorNet-B | 7 | 84.2 | 15.6G | 87M | | RMT-T | ViT | 82.4 | 2.5G | 14M |
| ConvFormer-S18 | 7 | 83.0 | 3.9G | 27M | | TransNeXt-M | ViT | 82.5 | 2.7G | 13M |
| ConvFormer-S36 | 7 | 84.1 | 7.6G | 40M | | RaCNN-T | CNN | **82.9** | 2.4G | 19M |
| ConvNeXt-T-dcls | 17 | 82.5 | 5.0G | 29M | | VMamba-T | Mamba | 82.2 | 4.5G | 22M |
| ConvNeXt-S-dcls | 17 | 83.7 | 9.5G | 50M | | SiMBA-B | Mamba | 82.6 | 5.5G | 27M |
| ConvNeXt-B-dcls | 17 | 84.1 | 16.5G | 89M | | CycleMLP-T | MLP | 81.3 | 4.4G | 28M |
| ConvNeXt-T-1D++ | 31 | 82.7 | 4.7G | 29M | | AS-MLP-T | MLP | 81.3 | 4.4G | 28M |
| ConvNeXt-B-1D++ | 31 | 83.8 | 15.8G | 90M | | ATMNet-T | MLP | 82.0 | 4.0G | 27M |
| VAN-B2 | 21 | 82.8 | 5.0G | 27M | | Wave-MLP-S | MLP | 82.6 | 4.5G | 30M |
| VAN-B3 | 21 | 83.9 | 9.0G | 45M | | RaMLP-T | MLP | 82.9 | 4.2G | 25M |
| VAN-B4 | 21 | 84.2 | 12.2G | 60M | | Swin-T | ViT | 81.3 | 4.5G | 29M |
| FocalNet-T | 3,5,7 | 82.3 | 4.5G | 29M | | HiViT-T | ViT | 82.1 | 4.6G | 18M |
| FocalNet-S | 3,5,7 | 83.5 | 8.7G | 50M | | CSWin-T | ViT | 82.7 | 4.3G | 23M |
| FocalNet-B | 3,5,7 | 83.9 | 15.4G | 89M | | CETNet-T | ViT | 82.7 | 4.3G | 23M |
| InceptionNeXt-T | 3,11 | 82.3 | 4.2G | 28M | | SG-Former-S | ViT | 83.2 | 4.8G | 23M |
| InceptionNeXt-S | 3,11 | 83.5 | 8.4G | 49M | | NAT-T | ViT | 83.2 | 4.3G | 28M |
| InceptionNeXt-B | 3,11 | 84.0 | 14.9G | 87M | | RaCNN-S | CNN | **83.9** | 4.2G | 28M |
| SLaK-T | 5,51 | 82.5 | 5.0G | 30M | | VMamba-S | Mamba | 83.6 | 8.7G | 50M |
| SLaK-S | 5,51 | 83.8 | 9.8G | 55M | | SiMBA-L | Mamba | 83.8 | 8.7G | 42M |
| SLaK-B | 5,51 | 84.0 | 17.1G | 95M | | CycleMLP-S | MLP | 82.9 | 8.5G | 50M |
| PeLK-T | 13,47,49,51 | 82.6 | 5.6G | 29M | | AS-MLP-S | MLP | 83.1 | 8.5G | 50M |
| PeLK-S | 13,47,49,51 | 83.9 | 10.7G | 50M | | ATMNet-B | MLP | 83.5 | 10.1G | 52M |
| PeLK-B | 13,47,49,51 | 84.2 | 18.3G | 89M | | Wave-MLP-B | MLP | 83.6 | 10.2G | 63M |
| UniRepLKNet-N | 3,5,7 | 81.6 | 2.8G | 18M | | Swin-S | ViT | 83.0 | 8.7G | 50M |
| UniRepLKNet-T | 3,5,7 | 83.2 | 4.9G | 31M | | HiViT-S | ViT | 83.5 | 9.1G | 38M |
| UniRepLKNet-S | 3,5,7 | 83.9 | 9.1G | 56M | | DAT-S | ViT | 83.7 | 9.0G | 50M |
| MogaNet-S | 3,5,7 | 83.4 | 5.0G | 25M | | BiFormer-B | ViT | 84.3 | 9.8G | 57M |
| MogaNet-B | 3,5,7 | 84.3 | 9.9G | 44M | | RaCNN-B | CNN | **84.5** | 8.7G | 51M |
| RaCNN-T | global | 82.9 | 2.4G | 19M | | VMamba-B | Mamba | 83.9 | 15.4G | 89M |
| RaCNN-S | global | 83.9 | 4.2G | 28M | | ATMNet-L | MLP | 83.8 | 12.3G | 76M |
| RaCNN-B | global | 84.5 | 8.7G | 51M | | RaMLP-B | MLP | 84.1 | 12.0G | 58M |
| RaCNN-B† | global | 85.0 | 11.4G | 51M | | NAT-B | ViT | 84.3 | 13.7G | 90M |
| | | | | | | RaCNN-B† | CNN | **85.0** | 11.4G | 51M |

Table 2: **(a) Comparison with CNN-based models on ImageNet-1K image classification. (b) Comparison with SOTA models on ImageNet-1K image classification.** All models are trained with the input resolution of $224 \times 224$, except † with $256 \times 256$.

| Models | Family | Reference | Top-1 | FLOPs | Params | Top-1 | FLOPs | Params |
|---|---|---|---|---|---|---|---|---|
| FastViT | ViT | ICCV23 | 75.6 | 0.7G | 4M | 79.1 | 1.4G | 7M |
| FAT | ViT | NeurIPS23 | 77.6 | 0.7G | 5M | 80.1 | 1.2G | 8M |
| SwiftFormer | ViT | ICCV23 | 78.5 | 1.0G | 6M | 80.9 | 1.6G | 12M |
| FasterNet | CNN | CVPR24 | 76.2 | 0.9G | 8M | 78.9 | 1.9G | 15M |
| MogaNet | CNN | ICLR24 | 77.2 | 1.0G | 3M | 80.0 | 1.4G | 5M |
| StarNet-S3 | CNN | CVPR24 | 77.3 | 0.8G | 6M | 78.4 | 1.1G | 8M |
| EfficientMod | CNN | ICLR24 | 78.3 | 0.8G | 7M | 81.0 | 1.4G | 13M |
| RepViT-M1 | CNN | CVPR24 | 78.5 | 0.8G | 5M | 80.6 | 1.3G | 8M |
| UniRepLKNet-F | CNN | CVPR24 | 78.6 | 0.9G | 6M | 80.2 | 1.6G | 11M |
| **RaCNN** | CNN | Our | **80.2** | 0.8G | 10M | **81.8** | 1.4G | 13M |

Table 3: **Comparison with lightweight models on ImageNet-1K image classification.** RaCNN-P and RaCNN-N are compared with other lightweight models with less than 1G FLOPs and with more than 1G FLOPs, respectively.

+0.8% Top-1 accuracy with similar or lower computation compared to the previous best ViT-based models (SwiftFormer (Shaker et al., 2023)) and CNN-based models (EfficientMod (Ma et al., 2024)).

### 4.2 OBJECT DETECTION

**Settings.** We conduct object detection experiments using RetinaNet (Lin et al., 2020) on the COCO (Lin et al., 2014) dataset. We follow the settings of Swin Transformer (Liu et al., 2021).

**Results.** We classify object detection baselines into two scales based on FLOPs, and the experimental results are presented in Table 4. RaCNN achieves leading performance in terms of $AP^b$ across

| Backbone | Family | Reference | $AP^b$ | $AP^b_{50}$ | $AP^b_{75}$ | $AP^b_S$ | $AP^b_M$ | $AP^b_L$ | Params | FLOPs |
|---|---|---|---|---|---|---|---|---|---|---|
| | | | RetinaNet (1× schedule) | | | | | | | |
| Swin-T | ViT | ICCV21 | 41.5 | 62.1 | 44.2 | 25.1 | 44.9 | 55.5 | 39M | 245G |
| CrossFormer++-S | ViT | TPAMI24 | 45.1 | 66.6 | 48.5 | 28.7 | 49.4 | 60.3 | 41M | 272G |
| WaveMLP-S | MLP | CVPR22 | 43.4 | 64.4 | 46.5 | 26.6 | 47.1 | 57.1 | 37M | 231G |
| ATMNet-S | MLP | AAAI23 | 43.6 | 64.9 | 46.8 | 27.2 | 47.5 | 57.9 | 37M | 233G |
| PlainMamba-Adapter-L1 | Mamba | BMVC24 | 41.7 | 62.1 | 44.4 | - | - | - | 19M | 250G |
| EfficientVMamba-B | Mamba | arXiv24 | 42.8 | 63.9 | 45.8 | 27.3 | 46.9 | 55.1 | 44M | - |
| MogaNet-S | CNN | ICLR24 | 45.8 | 66.6 | 49.0 | 29.1 | 50.1 | 59.8 | 35M | 253G |
| **RaCNN-S** | CNN | Our | **46.6** | **68.0** | **50.3** | **31.2** | **50.9** | **60.3** | 33M | 236G |
| Swin-S | ViT | ICCV21 | 44.7 | 65.9 | 49.2 | - | - | - | 98M | 477G |
| CrossFormer++-B | ViT | TPAMI24 | 46.6 | 68.4 | 50.1 | 31.3 | 50.8 | 61.5 | 62M | 389G |
| WaveMLP-B | MLP | CVPR22 | 44.2 | 65.1 | 47.1 | 27.1 | 47.8 | 58.9 | 66M | 334G |
| ATMNet-B | MLP | AAAI23 | 45.6 | 67.2 | 48.9 | 28.9 | 49.6 | 60.5 | 62M | 359G |
| VanillaNet-13 | CNN | NeurIPS23 | 43.0 | 62.8 | 44.3 | - | - | - | 75M | 397G |
| MogaNet-B | CNN | ICLR24 | 47.7 | 68.9 | 51.0 | 30.5 | 52.2 | 61.4 | 54M | 355G |
| **RaCNN-B** | CNN | Our | **47.8** | **68.9** | **51.6** | **31.4** | **52.2** | **61.5** | 57M | 327G |

Table 4: **COCO val2017 object detection results using various backbones employing a 1× training schedule.** FLOPs are evaluated with a resolution of $1280 \times 800$.

| Backbone | Family | Reference | $AP^b$ | $AP^b_{50}$ | $AP^b_{75}$ | $AP^m$ | $AP^m_{50}$ | $AP^m_{75}$ | Params | FLOPs |
|---|---|---|---|---|---|---|---|---|---|---|
| | | | Mask R-CNN (1× schedule) | | | | | | | |
| Swin-T | ViT | ICCV21 | 43.7 | 66.6 | 47.7 | 39.8 | 63.3 | 42.7 | 48M | 264G |
| CrossFormer++-S | ViT | TPAMI24 | 46.4 | 68.8 | 51.3 | 42.1 | 65.7 | 45.4 | 43M | 287G |
| SMT | ViT | ICCV23 | 47.8 | 69.5 | 52.1 | 43.0 | 66.6 | 46.1 | 40M | 265G |
| Hire-MLP-S | MLP | CVPR22 | 42.8 | 65.0 | 46.7 | 39.3 | 62.0 | 42.1 | 43M | 238G |
| ATMNet-T | MLP | AAAI23 | 44.8 | 66.9 | 49.0 | 41.0 | 64.2 | 44.3 | 47M | 251G |
| Vim-S-F | Mamba | arXiv24 | 43.1 | 65.2 | 47.3 | 39.3 | 62.2 | 42.3 | 44M | 272G |
| LocalVMamba-T | Mamba | arXiv24 | 46.7 | 68.7 | 50.8 | 42.2 | 65.7 | 45.5 | 45M | 291G |
| MogaNet-S | CNN | ICLR24 | 46.7 | 68.0 | 51.3 | 42.2 | 65.4 | 45.5 | 45M | 272G |
| **RaCNN-S** | CNN | Our | **48.0** | **70.0** | **52.7** | **43.3** | **67.0** | **46.7** | 43M | 254G |
| Swin-S | ViT | ICCV21 | 46.5 | 68.7 | 51.3 | 42.1 | 65.8 | 45.2 | 69M | 354G |
| CrossFormer++-S | ViT | TPAMI24 | 47.7 | 70.2 | 52.7 | 43.2 | 67.3 | 46.7 | 72M | 408G |
| SMT | ViT | ICCV23 | 49.0 | 70.2 | 53.7 | 44.0 | 67.6 | 47.4 | 52M | 328G |
| Hire-MLP-B | MLP | CVPR22 | 45.2 | 66.9 | 49.3 | 41.0 | 64.0 | 44.2 | 68M | 317G |
| ATMNet-B | MLP | AAAI23 | 46.5 | 68.6 | 51.0 | 42.5 | 66.1 | 45.8 | 72M | 377G |
| SiMBA-S | Mamba | arXiv24 | 46.9 | 68.6 | 51.7 | 42.6 | 65.9 | 45.8 | 60M | 372G |
| LocalVMamba-S | Mamba | arXiv24 | 48.4 | 69.9 | 52.7 | 43.2 | 66.7 | 46.5 | 69M | 414G |
| MogaNet-B | CNN | ICLR24 | 49.0 | 70.4 | 53.7 | 43.8 | 67.4 | 47.4 | 63M | 373G |
| **RaCNN-B** | CNN | Our | **49.1** | **70.9** | **53.7** | **44.1** | **68.0** | **47.5** | 66M | 346G |

Table 5: **COCO val2017 instance segmentation results using various backbones employing a 1× training schedule.** FLOPs are evaluated with a resolution of $1280 \times 800$.

different types of backbones in both scales. RaCNN surpasses the previous state-of-the-art CNN, MogaNet (Li et al., 2024) by 0.8% and 0.1% in $AP^b$ for the two scales, respectively, while having fewer FLOPs. RaCNN also significantly outperforms the well-known ViT-based backbone, Swin Transformer (Liu et al., 2021), by 5.1% and 3.1% in each group. Among smaller backbones, RaCNN leads CrossFormer++ (Wang et al., 2024), ATMNet (Wei et al., 2023) and EfficientVMamba (Pei et al., 2024) by margins of 1.5%, 3.0% and 4.1%, respectively. For larger backbones, RaCNN exceeds CrossFormer++ and ATMNet by 1.2% and 2.2%.

### 4.3 INSTANCE SEGMENTATION

**Settings.** Instance segmentation experiments are implemented with Mask R-CNN (He et al., 2020) and conducted on the COCO (Lin et al., 2014) dataset, also following the settings of Swin Transformer (Liu et al., 2021).

**Results.** Different models are grouped into two scales based on FLOPs, and the results are presented in Table 5. RaCNN surpasses all other models across all scales, exhibiting its powerful capability in instance segmentation. Specifically, for smaller models, RaCNN outperforms the state-of-the-art ViT SMT (Lin et al., 2023) by 0.2%, the state-of-the-art MLP ATMNet (Wei et al., 2023) by 3.2%, the state-of-the-art Mamba LocalVMamba (Huang et al., 2024) by 1.3%, and the state-of-the-art CNN MogaNet (Li et al., 2024) by 1.3%. Compared with larger backbones, RaCNN leads SMT, ATMNet, LocalVMamba and MogaNet by 0.1%, 2.6%, 0.7% and 0.1%. Additionally, RaCNN enjoys lower computational cost, simultaneously realizing high performance alongside high efficiency.

| Backbone | Family | Reference | mIoU | MS mIoU | Params | FLOPs |
|---|---|---|---|---|---|---|
| Swin-T | ViT | ICCV21 | 44.5 | 45.8 | 60M | 945G |
| AS-MLP-T | MLP | ICLR22 | - | 46.5 | 60M | 937G |
| CycleMLP-T | MLP | ICLR22 | - | 47.1 | 60M | 937G |
| EfficientVMamba-B | Mamba | arXiv24 | 46.5 | 47.3 | 65M | 930G |
| LocalVMamba-T | Mamba | arXiv24 | 47.9 | 49.1 | 57M | 970G |
| SLaK-T | CNN | ICLR23 | 47.6 | - | 64M | 957G |
| PeLK-T | CNN | CVPR24 | 48.1 | - | 62M | 970G |
| InceptionNeXt-T | CNN | CVPR24 | - | 47.9 | 56M | 933G |
| **RaCNN-S** | CNN | Our | **48.2** | **49.4** | 53M | 929G |
| Swin-S | ViT | ICCV21 | 47.6 | 49.5 | 81M | 1038G |
| AS-MLP-S | MLP | ICLR22 | - | 49.2 | 81M | 1024G |
| CycleMLP-S | MLP | ICLR22 | - | 49.6 | 81M | 1024G |
| SiMBA-S | Mamba | arXiv24 | 49.0 | 49.6 | 62M | 1040G |
| LocalVMamba-S | Mamba | arXiv24 | 50.0 | 51.0 | 81M | 1095G |
| SLaK-S | CNN | ICLR23 | 49.4 | - | 89M | 1057G |
| PeLK-S | CNN | CVPR24 | 49.6 | - | 84M | 1077G |
| InceptionNeXt-S | CNN | CVPR24 | - | 50.0 | 78M | 1020G |
| **RaCNN-B** | CNN | Our | **50.1** | **51.2** | 77M | 1025G |

Table 6: **The semantic segmentation results of different backbones on the ADE20K validation set with UperNet.** FLOPs are evaluated with a resolution of $2048 \times 512$.

| K | Top1 | FLOPs | Params |
|---|---|---|---|
| 3 | 82.9 | 2.4G | 19M |
| 5 | 82.8 | 2.4G | 19M |
| 7 | 82.8 | 2.5G | 20M |
| 9 | 82.6 | 2.6G | 20M |

Table 7: **The impacts of the kernel size of other DWConv.**

| RaFFN | RaGLU | Top1 | FLOPs | Params |
|---|---|---|---|---|
| ✗ | ✗ | 82.0 | 2.1G | 19M |
| ✓ | ✗ | 82.5 | 2.4G | 21M |
| ✗ | ✓ | 82.6 | 2.5G | 20M |
| ✓ | ✓ | 82.9 | 2.4G | 19M |

Table 8: **The impacts of the components.**

## 4.4 SEMANTIC SEGMENTATION

**Settings.** To evaluate the potential of RaCNN in semantic segmentation, we implement Uper-Net (Xiao et al., 2018) equipped with our RaCNN, and conduct experiments on the ADE20K (Zhou et al., 2017) dataset, following the settings of InceptionNeXt (Yu et al., 2024).

**Results.** Table 6 presents the semantic segmentation results. Among smaller backbones, RaCNN again excels beyond all other models w.r.t. both mIoU and MS mIoU. For larger backbones, RaCNN outperforms Swin (Liu et al., 2021), CycleMLP (Chen et al., 2022) and LocalVMamba (Huang et al., 2024). RaCNN also maintains its lead among other large-kernel CNNs (Liu et al., 2023; Chen et al., 2024; Yu et al., 2024) while requiring lower computational costs.

## 4.5 ABLATION STUDY

In this section, we utilize RaCNN-T to verify the effectiveness of the proposed components by conducting extensive ablation studies.

**Study on Kernel Size.** We increase the kernel size of traditional DWConv in the model and find that it negatively affects the results. We believe that since RaCNN has captured global information, using large kernel size in DWConv will introduce extra noises.

**Study on Components.** We replace RPWConv with DWConv in RaFFN and substitute RaGLU with FFN, and the loss of performance verify that all the proposed components have obvious effects.

## 5 CONCLUSION

This paper introduces the Region-aware CNN (RaCNN), which achieves a global receptive field without requiring extra techniques, yet surpasses state-of-the-art CNNs and ViTs. Specifically, we design the Region-aware Feed Forward Network (RaFFN) and Region-aware Gated Linear Unit (RaGLU) to capture global visual dependencies. The core of RaFFN is RPWConv, which divides spatial feature maps into several sparse global regions and generates dynamic weights within these regions, to capture coarse-grained global spatial cues. The RaCNN outperforms state-of-the-art CNNs, MLPs, ViTs, and Mambas in vision recognition, object detection, instance segmentation, and semantic segmentation while requiring fewer FLOPs.

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

## A APPENDIX

You may include other additional sections here.

