# OpenReview forum: "RaCNN: Region-aware Convolutional Neural Network with Global Receptive Field"
_ICLR.cc/2025/Conference — Submitted to ICLR 2025_

### Official Review · Reviewer_PZ92 · 2024-10-28

**Soundness:** 3
**Presentation:** 2
**Contribution:** 2
**Rating:** 5
**Confidence:** 4

**Summary:**

This paper proposes a novel convolutional neural network (CNN) model called Region-aware CNN (RaCNN), designed to simulate a large receptive field with minimal computational overhead. RaCNN achieves this by replacing the standard feedforward neural network (FNN) with RaGLU and the original depthwise convolution (DWConv) with Region PWConv (RPWConv), allowing the model to capture long-range visual dependencies. The paper demonstrates the effectiveness of RaCNN through experiments on classification, detection, and segmentation tasks.

**Strengths:**

The motivation of the paper is valuable. The high computational complexity of large kernel size is indeed a barrier in the field of computer vision.

The proposed RaCNN model shows effectiveness across various tasks, as evidenced by the experimental results.

**Weaknesses:**

The contribution of this work is not particularly groundbreaking. Compared to previous methods like [], the advantages of RaCNN are not significantly clear.

Some expressions in the paper lack precision. For example, the distinction between “FLOPS” and “FLOPs” on page 8, line 421, is not handled correctly. I recommend a thorough review of the manuscript.

Equation (6) appears to be incorrect; “x·x” cannot be computed as written. Please verify all equations.

**Questions:**

The paper lacks detailed explanation regarding the advantages of dilated windows in Region PWConv. Could you provide more clarification on this?

What would be the impact of using Dynamic PWConv instead of Region PWConv? Additional ablation studies would strengthen the validation of your method.

In my view, Figure 2(e) does not effectively demonstrate RaCNN’s ability to capture long-range dependencies.

In Table 8, row 4 (which uses both ReFFN and RaGLU) reports fewer FLOPs and parameters than rows 2 and 3, which is puzzling. Could you explain this discrepancy?

---

> ### Author Response · Authors · 2024-11-25
> **Response to Reviewer PZ92**
>
> Thank you for your valuable time and insightful comments!
>
> **[W1] Limited contribution**
>
> We believe our proposed method introduces novelty:
>
> + RPWConv is a new kind of dynamic CNN that generates weights dynamically for different inputs and positions, providing a global receptive field.
> + The Region Attention operation in RaGLU solves the serious information loss problem caused by global average pooling, and utilizes sparse global pooling to capture long-range dependencies.
>
> Our method also introduces great performance gains:
>
> + RaCNN delivers more significant performance gains than previous methods that generally showed limited improvements around 0.3%, as displayed in Table 1.
> + In lightweight structures, our RaCNN shows even greater performance gains. For example, compared with the state-of-the-art UniRepLKNet [1], RaCNN-P achieves a significant 1.6% accuracy improvement (80.2% vs. 78.6%).
> + Please refer to the response to `[W1]` of Reviewer `pPeK` for detailed explanations about the performance gains.
>
>
>
> **[W2] Imprecise expressions such as the distinction between “FLOPS” and “FLOPs”**
>
> Thank you for pointing out this typo, and we have corrected it. We have also carefully reviewed our manuscript again to eliminate other typos.
>
>
>
> **[W3] Incorrectness in Equation (6)**
>
> We have modified $x\cdot x$ to $xx^T$ in Equation (6), since $x\in R^{c\times hw}$ represents a matrix rather than a vector. We have also carefully checked all equations to guarantee their correctness.
>
>
>
> **[Q1] Detailed clarification regarding the advantages of dilated windows in Region PWConv**
>
> In Region PWConv, we divide a feature map into into $\frac{H}{rs}\times\frac{W}{rs}$ spatial regions ($H$ and $W$ are the height and width of the map, and $rs$ is the region size). Then we sample one position in each region, and combine all these sampled positions into a vector in each channel. In another word, positions with the same color in Figure 4(c) form a dilated window containing sparse global information, and each dilated window in each channel forms a vector. Dynamic weights are generated in each dilated window, allowing for efficient and flexible weight generation.
>
> Based on the above detailed explanation of Region PWConv, we conclude its advantages as follows:
>
> + **Sparse global perception:** Each dilated window contains sparse global information, so Region PWConv can capture global cues.
> + **Dynamic region-aware weights:** Region PWConv can dynamically generate region-aware weights. Vanilla PWConv generates static weights for different inputs. Dynamic PWConv improves it by generating dynamic weights for different input, but all positions share the same weights. Region PWConv not only tailors dynamic weights for different inputs, but also provides different weights across different dilated windows.
> + **Linear complexity:** Region PWConv has a linear complexity w.r.t. the size of feature map, which is more efficient than quadratic complexity self-attention. The proof can be found in the response to `[Q1]` of Reviewer `EQAG`.
>
> Note that we will modify the term "dilated window" to "sparse global window" for clarity in edited version of PDF.
>
>
>
> **[Q2] Impact of using Dynamic PWConv instead of Region PWConv**
>
> Dynamic PWConv generates the same weights shared across all positions. This approach causes some background noise to be weighted in the same way as important regions, interfering with the learning of crucial information. Region PWConv generates different weights for different positions to solve this problem.
>
>
>
> **[Q3] Figure 2(e) does not effectively demonstrate RaCNN’s ability to capture long-range dependencies**
>
> Figure 2(e) shows that our RaCNN has greater receptive weights in the central region, but its receptive field also covers edge areas. Notice that the receptive weights are larger in the bottom-left corner, indicating that RaCNN can cover the entire image.
>
> The visualization is performed on models trained on ImageNet. For ImageNet, the important areas are mainly in the center. Baselines like SLaK [2] and InceptionNeXt [3] overly focus on edge areas, leading to performance degradation on ImageNet.
>
> For more details about Figure 2, please refer to `[Common Question 2]`.
>
>
> **[Q4] Explain why row 4 in Table 8 reports fewer FLOPs and parameters than rows 2 and 3**
>
> Without using the FFN, our method is equivalent to the Channel MLP used in Swin Transformer [4]. This module incurs higher computational costs and cannot capture spatial information, resulting in poorer performance.
>
> ---
>
> **Reference**
>
> [1] UniRepLKNet: A Universal Perception Large-Kernel ConvNet for Audio, Video, Point Cloud, Time-Series and Image Recognition. CVPR 2024
> [2] More ConvNets in the 2020s: Scaling up Kernels Beyond 51x51 using Sparsity. ICLR 2023
> [3] InceptionNeXt: When Inception Meets ConvNeXt. CVPR 2024
> [4] Swin Transformer: Hierarchical Vision Transformer using Shifted Windows. ICCV 2021

---

### Official Review · Reviewer_zCMe · 2024-11-03

**Soundness:** 3
**Presentation:** 3
**Contribution:** 3
**Rating:** 5
**Confidence:** 3

**Summary:**

This paper presents RaCNN, a convolutional neural network designed to capture both long-range dependencies and local contextual information in images. The architecture introduces two key components: the Region-aware Feed Forward Network (RaFFN) and the Region-aware Gated Linear Unit (RaGLU). RaFFN employs a specialized Region Point-Wise Convolution (RPWConv), which divides spatial feature maps into global regions and applies dynamic weights within each, allowing for the capture of both global and local features. RaGLU enhances spatial information mixing through region-specific attention mechanisms. The authors claim that RaCNN outperforms state-of-the-art models such as MogaNet and Swin Transformer in tasks like image classification, object detection, and instance segmentation, while maintaining computational efficiency. Extensive experiments and ablation studies are provided to validate RaCNN's effectiveness.

**Strengths:**

RaFFN and RaGLU introduce a novel region-based dynamic weighting approach in CNNs, allowing RaCNN to capture multiscale spatial dependencies compared to traditional convolutional methods more effectively.

The experimental results across multiple vision tasks, such as object detection and instance segmentation, demonstrate RaCNN's robustness. This approach efficiently captures global dependencies while maintaining computational efficiency, representing an advancement over previous CNNs and ViTs.

RaCNN demonstrates scalability across various model sizes and tasks.

**Weaknesses:**

The claim that RaCNN "can capture long-range dependencies and local context features simultaneously without excessive noise" is based on a single ERF visualization. However, visualizations from a single layer are insufficient to support such broad conclusions about noise reduction and contextual feature capture, as ERF patterns vary across layers in complex networks. A more comprehensive analysis across multiple layers or the use of additional metrics would be necessary to fully substantiate this claim.

Although the paper highlights RaCNN’s computational efficiency, it lacks a detailed comparison of how this efficiency holds up against similar models in real-world scenarios, such as inference time or memory usage.

The RPWConv design, which divides the spatial feature map into predefined regions, may limit flexibility across different image types. Fixed partitioning could reduce generalizability for diverse visual tasks that benefit from more adaptive receptive fields. Further investigation into alternative partitioning strategies, like adaptive region sizes, would strengthen the proposed approach.

While the ablation studies focus on the impact of individual components (e.g., RPWConv, RaGLU), they do not sufficiently explore the interactions between these modules. Expanding the ablation studies to analyze how these components interact could provide deeper insights into RaCNN’s overall design.

The paper’s focus on FLOPs as the primary metric for efficiency may overlook other crucial factors, such as memory usage and inference time. These metrics are critical for real-world deployment, and their inclusion would provide a more balanced perspective on RaCNN’s efficiency.

**Questions:**

1. Could the authors provide additional evidence to support RaCNN’s claims of noise resistance and contextual feature capture? Specifically, how does the ERF visualization change across different layers of RaCNN, and would deeper layer results reinforce the conclusions drawn from Figure 2?

2. How does RaCNN perform in terms of inference speed and memory usage compared to models like Swin Transformer and MogaNet in real-time applications?

3. Have the authors considered exploring alternative region partitioning strategies for RPWConv, such as using adaptive region sizes, to enhance flexibility across different image types?

---

> ### Author Response · Authors · 2024-11-25
> **Response to Reviewer zCMe**
>
> We express our earnest gratitude for your valuable time and feedback!
>
>
>
> **[W1, Q1] Visualizations of ERF limited in a single layer are not sufficient for conclusions about noise reduction and contextual feature capture**
>
> We utilized the visualization tool provided by RepLKNet [1], which analyzes the **complete network** and its corresponding pre-trained weights, rather than focusing on individual layers. Therefore, this method does not support single-layer analysis. We apologize for any confusion caused by this limitation. In future, we will train multiple models with fewer layers to further analyze their effects at the layer level.
>
>
>
> **[W2] Lack of comparison of efficiency with baselines in real-world scenarios**
>
> In the Throughput column of Table 1, we compare the training speeds of RaCNN and other CNNs, which addresses your concern. Specifically, compared to SLaK [2], which uses $51\times 51$ kernels, our RaCNN-T achieves up to **700% throughput improvement** (3037 img/s vs. 417 img/s). Compared to RepLKNet [1] applying $31\times 31$ kernels, our RaCNN-S achieves up to **370% throughput improvement** (2185 img/s vs. 585 img/s). These experimental results emphasize the efficiency advantage of RaCNN in real-world scenarios.
>
>
>
> **[W3, Q3] Fixed partitioning in RPWConv may limit flexibility, and exploration of alternative region partitioning strategies such as adaptive region size**
>
> The region size is set to 7, following the design of classic models like Swin Transformer [3] and ConvNeXt [4]. Further increasing the region size yields almost no performance improvement, while reducing the size to 4 results in a 0.1% to 0.2% drop in performance on ImageNet.
>
> As for alternative region partitioning strategies, we experimented with adaptive region sizes. Unfortunately, this strategy significantly degrades the training speed, causing **nearly a 200% slowdown** under the same FLOPs. We suspect this phenomenon is due to memory access inefficiencies.
>
>
> **[W4] Insufficient ablation studies on the interactions between components**
>
> We appreciate your feedback and will include more ablation studies on the interactions between components in future versions. Due to time, article length and hardware constraints, we can only provide the ablation experiments in the current version.
>
> For additional ablation studies:
>
> + If you are interested in the impact of cosine distance vs. inner-product in Dynamic PWConv, please refer to the response to` [W2]` of Reviewer `pPeK`.
>
> + If you are interested in the impact of region size and region partitioning strategy, please refer to the response to `[W3,Q3]`.
>
>
>
> **[W5] Efficiency metric FLOPs overlooks crucial factors such as memory usage and inference time in real-world deployment**
>
> Besides FLOPs, we also provide the training throughput in Table 1 as an important efficiency metric. For detailed analysis, please refer to the response to `[W4]`.
>
> To provide a more comprehensive analysis of real-world scenarios, we will include experimental results on model inference speed based on the CoreML framework on iPhone 12 and the TensorRT framework on Nvidia 4090 in future versions.
>
>
>
> **[Q2] Inference speed and memory usage of RaCNN compared to models like Swin Transformer and MogaNet in real-time applications**
>
> We will include comparison experiments in terms of inference speed and memory usage on the CoreML framework on iPhone 12 and the TensorRT framework on Nvidia 4090 in future versions. Due to time, article length and hardware constraints, we can only provide the experiments in the current version.
>
> However, based solely on the training throughput provided in Table 1, our RaCNN is **nearly 100% faster than MogaNet**, which gives us reason to infer that RaCNN's inference speed will also have a significant advantage over MogaNet, and that RaCNN is promising in real-time applications.
>
> ---
>
> **Reference**
>
> [1] Scaling Up Your Kernels to 31×31: Revisiting Large Kernel Design in CNNs. CVPR 2022
> [2] More ConvNets in the 2020s: Scaling up Kernels Beyond 51x51 using Sparsity
> [3] Swin Transformer: Hierarchical Vision Transformer using Shifted Windows. ICCV 2021
> [4] A ConvNet for the 2020s. CVPR 2022

---

> > ### Comment · Reviewer_zCMe · 2024-11-25
> >
> > Thank you for your detailed response. However, given the limited technical contribution and the manuscript's current presentation, I will maintain my original evaluation score.

---

### Official Review · Reviewer_CKsH · 2024-11-03

**Soundness:** 2
**Presentation:** 1
**Contribution:** 2
**Rating:** 3
**Confidence:** 5

**Summary:**

This paper presents the RaCNN for vision recognition. The proposed RaCNN achieves a global receptive field with two designs: the region-aware FFN that uses a variant of dilated attention to model global relations, and the region-aware GLU that uses DWConv and a regional SE module to improve model capacity. The model is evaluated on widely used visual recognition benchmarks and compared to recent SoTA methods.

**Strengths:**

- The method achieves competitive performance on various visual recognition tasks including classification and dense prediction. The throughput also looks good.

- The authors conducted extensive experiments to verify the model.

**Weaknesses:**

- The proposed model actually is not a convolutional neural network. A convolution operation should be transition-equivariant. The proposed operation is closer to a variant of local attention instead of convolution. The authors may consider changing the name of the model and some claims in the paper to avoid misunderstanding.

- The proposed design is a mixture of some existing techniques.  The Region Point-Wise Convolution is not a convolutional operation and it is very similar to [r1]. The RA operation is a variant of the SE module. Although the paper shows that the mixture can achieve good performance, the technical contribution is quite limited.

[r1] Glance-and-Gaze Vision Transformer, NeurIPS 2021.

- The motivation of the paper is not quite clear. According to the abstract, the paper wants to propose a new solution because "these models introduce specialized techniques like re-parameterization, sparsity, and weight decomposition, increasing the complexity of the training and inference stages". However, I think the proposed solution also introduces many complex operations and may make the solution less general.

- The presentation of the paper needs improvement. Many claims are not clear. For example, in the caption of Figure 1, it is mentioned that some existing methods "could capture long-range dependency but introduce excessive visual noises". I don't found there is evidence to show these methods can introduce extra noise. Besides, the descriptions of the core contribution: RPWConv and RA, are quite confusing. Figure 4 didn't clearly the RPWConv operation. Equations 5-7 are also misleading since both "x \dot x" and "xw" actually are matrix multiplication. I need to check the provided code to clearly understand how the operation actually works.

**Questions:**

Please refer to my comments above. Considering there are many issues about the presentation, positioning, technical contribution, and motivation, I cannot recommend acceptance for this paper.

---

> ### Author Response · Authors · 2024-11-25
> **Response to Reviewer CKsH (part 1)**
>
> We are sincerely grateful for dedicating your time and effort to review our work!
>
> **[W1] Doubt that the proposed model is not a convolutional neural network**
>
> Transition-equivariance is a property of the **classic** CNNs. Nevertheless, CNNs are not necessarily transition-equivariant, as dynamic CNNs such as [1] [2] [3] [4] lack this property. In this context, **our RaCNN is also a type of dynamic CNNs**.
>
> We admit that the proposed RPWConv and RaGLU are variants of Channel Attention. However, **we disagree with the view that convolution and attention are mutually exclusive**. In fact, we think Channel Attention is a technique that integrates the properties of both convolution and attention. Under this understanding, **previous methods, such as SENet [5], CBAM [6] and SKNet [7], have already incorporated attention mechanisms into CNNs**. We identify their shortcomings—they all use vanilla pooling to compress information, leading to great information loss. To address this, RPWConv uses sparse global windows to retain all information for a more complete representation, and Region Attention utilizes sparse global pooling to preserve global cues.
>
> Moreover, RPWConv can be **mathematically formulated in the same way as PWConv.** For vanilla PWConv, its weights $w\in R^{c_{\text{out}}\times c_{\text{in}}}$ (i.e. the kernel parameters) are static, and it can be formulated as
>
> $y=wx$
>
> where $x\in R^{c_\text{in}\times hw}$ and $y\in R^{c_\text{out}\times hw}$ represent input and output of PWConv, respectively. For Dynamic PWConv, $w$ is dynamically generated as described in Equation 6 of the article, and the output $y$ is obtained using Equation 7, which is same as the equation above. Similarly, RPWConv can also be formulated in this manner, with the improvement that $w$ is generated and shared for each sparse global window. This demonstrates that RPWConv adheres to the mathematical framework of a point-wise convolutional network.
>
> In summary, RaCNN is CNN-based. Thus, we assert that referring to our model as a “Convolutional Neural Network” in the title is both accurate and justified.
>
>
> **[W2] The proposed design is a mixture of some existing techniques**
>
> **RPWConv:**
>
> + From a conceptual standpoint, our RPWConv is fundamentally different from GG-Transformer [8] mentioned by the reviewer:
>
>   + **GG-Transformer is a spatial attention mechanism**, where each token corresponds to the vector at a single spatial position.
>
>   + In contrast, **RPWConv is a channel attention mechanism**. For Dynamic PWConv, each token corresponds to a channel. For the improved RPWConv, each token corresponds to multiple spatial positions within the same sparse global window in a channel — specifically, the multiple spatial positions with the same color (in Figure 4(c)) in a channel.
>
> + Furthermore, RPWConv is mathetically distinct from GG-Transformer:
>
>   + RPWConv can be formulated equivalently to PWConv, i.e. $y=wx$. For vanilla PWConv, $w$ is static. For RPWConv, $w$ is dynamically generated in a region-aware manner. Further details are provided in the response to `[W1]`.
>
>   + However, Transformer-based methods including GG-Transformer cannot be unified under this formula.
>
> **RA:** The RA operation in RaGLU is indeed a variant of the SE operation. However, RA addresses the criticism of SE for causing significant information loss due to global average pooling, which makes it valuable.
>
>
> **[W3] Unclear motivation about reducing training and inference complexity in previous methods**
>
> Table 1 demonstrates that other models using specialized techniques decrease the training speed. In contrast, our RaCNN improves performance without compromising training efficiency. For example, compared to SLaK [9], which uses dynamic sparsity to combine two strip convolutions, our RaCNN achieves a 0.4% performance improvement (82.9% vs 82.5%), while being **700% faster** in training speed.  Similarly, compared to RepLKNet [10], which applies re-parameterization, our RaCNN achieves a 0.4% performance gain (83.9% vs 83.5%) with **370% faster** training speed and only **35% of the parameters**. This indicates that our proposed method successfully handles the challenges in our motivation, balancing performance with training and inference efficiency
>
> As for concerns about increased complexity and reduced generalization, we want to emphasize that our RaCNN can be implemented with hardware-friendly operations. Specifically, the sparse global window in RPWConv and RA can be implemented simply with tensor reshaping operations without extra complexity. We also eliminate inefficient large kernels and only use highly-optimized $3\times3$ convolutions, further enhancing its scalability on hardwares.

---

> ### Author Response · Authors · 2024-11-25
> **Response to Reviewer CKsH (part 2)**
>
> **[W4] Several presentation problems**
>
> We thank the reviewer for listing the presentation problems. We will respond them one by one:
>
> 1. The reviewer mentioned that Figure 2 cannot prove that other baselines introduce extra noise.
>
>    In Figure 2, the ERF of SLaK [9] assigns excessive weight to the edges, which is unreasonable. Similarly, InceptionNeXt [11] also assigns excessive weight to strip-shaped regions due to its strip-shaped convolutions, introducing additional noise. In comparison, our RaCNN focuses more on the central region while assigning smaller weights to the edges, thereby avoiding the introduction of irrelevant information from edge areas. Note that in the ImageNet dataset, semantic regions are primarily concentrated in the center of images, and that's why center cropping is commonly used during testing. Refer to `[Common Question 2]` for further analysis of the visualization in Figure 2.
>
>    To enhance clarity, we will add more detailed explanations to the caption of Figure 2 in the revised version of PDF.
>
> 2. The description of RPWConv and RA is confusing.
>
>    We apologize for the confusion caused. We will provide updated diagrams of RPWConv and RA to illustrate our methods more clearly.
>
>    If you have questions about how RPWConv and RA achieve global receptive fields, please refer to `[Common Question 1]`. If you have questions about how RaCNN combines channel attention with CNN essence, please refer to the responses to `[W1]` and `[W2]`. If you are interested in the detailed explanations of the advantages of RPWConv, please refer to the response to `[Q1]` of Reviewer `PZ92`.
>
> 3. Equations 5-7 are misleading.
>
>    For clarity, we modify $x\cdot x$ to $xx^T$ in Equation 6, as $x$ represents a matrix rather than a vector. Then Equations 5-7 become
>    $$
>    x = \text{Reshape}(x^l) \in \mathbb{R}^{c \times hw} \tag{5}
>    $$
>
>    $$
>    w = \text{Softmax}\left(s \frac{xx^T}{\|x\|_2^2}\right) \tag{6}
>    $$
>
>    $$
>    y = \text{Reshape}(wx) \in \mathbb{R}^{c \times h \times w} \tag{7}
>    $$
>
>    We will also declare "$w\in R^{c\times c}$ is the generated dynamic weight" below these equations for better clarity.
>
>    Note that the matrix multiplication $wx$ is the definition of PWConv operation.
>
> ---
>
> **Reference**
>
> [1] Dynamic Convolution: Attention over Convolution Kernels. CVPR 2020
> [2] Dynamic Convolutions: Exploiting Spatial Sparsity for Faster Inference. CVPR 2020
> [3] Involution: Inverting the Inherence of Convolution for Visual Recognition. CVPR 2021
> [4] Dynamic Region-Aware Convolution. CVPR 2021
> [5] Squeeze-and-Excitation Networks. CVPR 2018
> [6] CBAM: Convolutional Block Attention Module. ECCV 2018
> [7] Selective Kernel Networks. CVPR 2019
> [8] Glance-and-Gaze Vision Transformer. NeurIPS 2021
> [9] More ConvNets in the 2020s: Scaling up Kernels Beyond 51x51 using Sparsity
> [10] Scaling Up Your Kernels to 31×31: Revisiting Large Kernel Design in CNNs. CVPR 2022
> [11] InceptionNeXt: When Inception Meets ConvNeXt. CVPR 2024

---

> > ### Comment · Reviewer_CKsH · 2024-11-29
> >
> > Thanks for the detailed reply. The reason I suggest another name is to avoid misunderstanding. Some presentation issues have been resolved but I think the paper still needs a thorough revision. After reading other reviews as well as the reply,  I still think the paper is not ready for publication now. Therefore, I would keep my initial rating.

---

### Official Review · Reviewer_EQAG · 2024-11-04

**Soundness:** 2
**Presentation:** 2
**Contribution:** 2
**Rating:** 5
**Confidence:** 3

**Summary:**

The paper titled "RaCNN: Region-Aware Convolutional Neural Network with Global Receptive Field" proposes a new CNN model, RaCNN, that enhances the receptive field to capture global visual dependencies without increasing computational complexity. The key innovations of RaCNN include two modules: the Region-aware Feed Forward Network (RaFFN) and the Region-aware Gated Linear Unit (RaGLU).

RaFFN uses Region Point-Wise Convolution (RPWConv) to capture global cues by dividing spatial feature maps into sparse global regions, allowing it to dynamically adjust weights per region. Meanwhile, RaGLU serves as a channel mixer, capturing long-range dependencies in a sparse, global manner, improving spatial information aggregation. Compared to conventional CNNs, RaCNN demonstrates state-of-the-art performance, surpassing models like MogaNet, while requiring fewer computational resources.

**Strengths:**

1.	The paper proposes region-aware CNN. The author suggest RaCNN captures both local and global information to better feature extraction
2.	Experiments show RaCNN achieves better results on various tasks, including image classification (ImageNet-1K), object detection (COCOval2017), instance segmentation (COCOval2017), and semantic segmentation (ADE20K).

**Weaknesses:**

1.	It is confused to figure out how RaCNN captures global information. For instance, Region PWConv in figure 4c takes dilated windows and Region Attention in figure 5 averages spatial feature in a dilated manner. However, dilation is not equivalent to the so-called ‘global’. From my point of view, it is more proper to say it captures a larger local receptive field.
2.	The author suggests RaCNN is better because it captures local and global information. However, in figure 2 we can see SLak, UniRepLKNet, and InceptionNeXT all captures local and global receptive field, even with a more ‘global’ field. Therefore, I am suspecting local and global receptive field is not the true reason for RaCNN performance.
3.	For figure 1, there is no margin between the figure 1 caption and main text.
4.	More tasks [1] should be added to verify the effectiveness of RaCNN, such as 2D and 3D human pose estimation, and video prediction.
[1] Li, Siyuan, Zedong Wang, Zicheng Liu, Cheng Tan, Haitao Lin, Di Wu, Zhiyuan Chen, Jiangbin Zheng, and Stan Z. Li. "Moganet: Multi-order gated aggregation network." In The Twelfth International Conference on Learning Representations. 2023.

**Questions:**

Question: While RaCNN claims to achieve a global receptive field without additional complexity, it would be beneficial to understand the exact computational trade-offs compared to recent models with large-kernel convolutions. Could the authors provide further detail on any specific optimizations that contribute to this efficiency?
Suggestion: Including a breakdown of computational complexity relative to specific design choices (like RPWConv and RaGLU) and a comparison against representative models would enhance clarity on how RaCNN maintains low computational cost.
Scalability Across Tasks and Architectures:

Question: RaCNN demonstrates strong performance across multiple vision tasks. How does its performance and efficiency vary when scaling to larger or smaller model variants, particularly in dense prediction tasks? Are there configurations where RaCNN’s advantages diminish?
Suggestion: It would be helpful if the authors provided more details on the model's performance when scaled to different architectures and task demands, particularly in cases where efficiency gains may be less pronounced.
Impact of Region Size and Sparse Global Regions:

Question: RaCNN’s RPWConv operates on sparse global regions, which raises questions about the impact of different region sizes on performance and the possibility of information loss. Could the authors clarify how they determine optimal region sizes and the robustness of RaCNN to varying region granularity?
Suggestion: An ablation study on the impact of different region sizes and sparsity levels on accuracy and computation might help clarify the balance between global context capture and computational efficiency.

---

> ### Author Response · Authors · 2024-11-25
> **Response to Reviewer EQAG (part 1)**
>
> We highly apreciate your time and effort in reviewing our submission!
>
> We firstly respond to your concerns about the weakness.
>
> **[W1] Doubt about global receptive field, and difference between dilation and "global"**
>
> Traditional dilated convolutions insert holes between kernel elements to expand the kernel, but the kernel remains local. In contrast, our method divides the entire feature map into $\frac{H}{rs}\times\frac{W}{rs}$ spatial regions ($H$ and $W$ are the height and width of the feature map, and $rs$ is the region size), and samples one position from each region. In another word, positions with the same color in Figure 4(c) form a sparse global window (called "dilated window" in the original PDF), and each window provides a global view of the image. This allows RaCNN to capture global information effectively.
>
> We apologize for the confusion. Renaming it to 'Global Sparse' would make it easier to understand than 'Dilated'.
>
> For more details about global receptive fields, please refer to `[Common Question 1]`.
>
> **[W2] Comparison of receptive field in Figure 2 cannot prove that RaCNN has local and global receptive field, and cannot explain RaCNN's performance.**
>
> SLaK [1], UniRepLKNet [2] and InceptionNeXt [3] have large receptive fields and can capture global information. However, their static weights are a major drawback, limiting their ability to handle diverse and complex inputs. In contrast, RPWConv in RaCNN dynamically generates weights tailored for input content, enhancing adaptability and achieving better performance.
>
> Additionally, these large-kernel CNNs all use large kernels ($\ge11$), whereas RaCNN only uses standard $3\times3$ convolutions with proposed novel techniques to simulate large kernels and achieve a global receptvie field. This design also offers significant speed advantages, as shown in Table 1.
>
> For more details about Figure 2, please refer to `[Common Question 2]`.
>
>
> **[W3] Formatting error in Figure 1**
>
> Thank you for pointing out this error, and we have corrected it. We have also carefully reviewed our manuscript again to ensure no other formatting issues remain.
>
>
>
> **[W4] More downstream tasks to verify the effectiveness of RaCNN**
>
> We appreciate your suggestion. Due to time constraints and the length limitation of the article, we reference the majority of related works to complete the detection and segmentation tasks. Additional experiments will be included in future versions to further validate RaCNN’s effectiveness.

---

> ### Author Response · Authors · 2024-11-25
> **Reponse to Reviewer EQAG (part 2)**
>
> In this part, we answer your questions.
>
> **[Q1] More details on specific optimizations contributing to RaCNN's efficiency**
>
> Both RPWConv and RaGLU can be highly optimized using mature techniques to achieve high efficiency.
>
> + RPWConv can be regarded as a sparse global channel self-attention. Therefore, we can directly leverage existing well-established attention algorithm, such as Efficient Attention [1] and FlashAttention [2], to optimize the model.
>
> + RaGLU is a variant of the Channel MLP. Its basic modules are also highly mature and widely adopted in large language models and can be efficiently optimized. Furthermore, its upsampling operation uses nearest-neighbor interpolation, and its downsampling operation is a pooling algorithm. Both operations benefit from well-established optimization techniques.
>
> Additionally, RPWConv and RaGLU have linear theoretical complexity w.r.t. input size, which is more efficient than vanilla self-attention with quadratic complexity.
>
> + Dynamic PWConv is a type of channel attention with complexity $O(hwc^2)$, where $c$, $h$ and $w$ are channels, height and width of the feature map, respectively. Assuming RWPConv has a region size of $rs$, it can be regarded as $rs^2$ Dynamic PWConvs in all sparse global windows, so its complexity is $rs^2\times O(\frac{h}{rs}\cdot\frac{w}{rs}\cdot c^2)=O(hwc^2)$, which is still linear to the input size.
> + In Region Attention, sparse global average pooling has complexity $rs^2\times O(\frac{h}{rs}\cdot\frac{w}{rs}\cdot c)=O(hwc)$. The two PWConvs have complexity $O(c\cdot r)+O(r\cdot c)=O(r\cdot c)$, where $r$ is the number of output channels in the first PWConv. Nearset-neighbor upsampling and elementwise multiplication both have complexity $O(hwc)$. Therefore, the total complexity is $O(hwc+rc)$, which is linear the to input size.
>
>
>
> **[Q2] How RaCNN's performance and efficiency vary across different variants, especially for dense prediction tasks, and whether there are configurations where its advantages diminish**
>
> We have already validated 5 variants of RaCNN (-P, -N, -T, -S and -B), and their effectiveness has been verified in dense prediction tasks such semantic segmentation and instance segmentation. The results of performance and efficiency are displayed in Table 2~6. Currently, we are working on training a 100M parameter large model; however, due to time constraints, we have not yet obtained the results.
>
> Our experiments show that RaCNN reaches leading performance and higher efficiency across various configurations. In lightweight settings, RaCNN offers a significant performance advantage (a 1.6% gain for RaCNN-P, and a 0.8% gain for RaCNN-N). In larger settings, RaCNN still delivers considerable performance improvements and speed advantages, securing its leading position across different tasks.
>
>
>
> **[Q3] How to determine optimial region sizes**
>
> For this, we follow the classic models like Swin Transformer [3] and ConvNeXt [4], and set the region size to 7. When the region size is increased further, there is almost no performance gain. However, reducing the region size to 4 results in a performance drop of 0.1% to 0.2% on ImageNet.
>
> ---
>
> **Reference**
>
> [1] Efficient Attention: Attention with Linear Complexities. WACV 2021
> [2] FlashAttention: Fast and Memory-Efficient Exact Attention with IO-Awareness. NIPS 2022
> [3] Swin Transformer: Hierarchical Vision Transformer using Shifted Windows. ICCV 2021
> [4] A ConvNet for the 2020s. CVPR 2022

---

### Official Review · Reviewer_pPeK · 2024-11-06

**Soundness:** 3
**Presentation:** 3
**Contribution:** 3
**Rating:** 5
**Confidence:** 5

**Summary:**

The paper introduces Region-aware CNN (RaCNN), a novel architecture that achieves a global receptive field without added complexity, outperforming state-of-the-art models. RaCNN features two key modules: the Region-aware Feed Forward Network (RaFFN) using Region Point-Wise Convolution (RPWConv) to capture global visual cues, and the Region-aware Gated Linear Unit (RaGLU) for capturing long-range dependencies. With only 84% of the computational complexity, RaCNN surpasses the performance of MogaNet (83.9% vs. 83.4%) and demonstrates strong scalability, excelling in object detection, instance segmentation, and semantic segmentation.

**Strengths:**

1. RaCNN includes two critical modules: the Region-aware Pointwise Convolution (RPWConv) for capturing global visual information and the Region-aware Gated Linear Unit (RaGLU) for capturing long-range dependencies.
2. RaCNN outperforms state-of-the-art models like MogaNet, achieving higher performance (83.9% vs. 83.4%) with only 84% of the computational complexity.
3.  The model demonstrates excellent performance across various tasks, including object detection, instance segmentation, and semantic segmentation.The paper provides a thorough comparison with other methods across multiple downstream tasks, showcasing the robustness and effectiveness of RaCNN.

**Weaknesses:**

1. While RaCNN shows improvements over existing models, the performance gains might be considered incremental.
2. The ablation experiments are insufficient, lacking further analysis on the effectiveness of the proposed modules. For example, the dynamic PWconv use cosine distance to measure similarity instead of inner-product, how about the difference affect the performance ?
3. From the definition of the dynamic PWconv and RaGLU in the paper, and its implementation in the code, the proposed method does not capture the global receptive field of imagesm which is inconsistent with the paper's title.  Since the dynamic PWconv are improvement upon the basic of Pointwise Convolution, the dynamic wieghts generated for different region just provide different ways for gatherring features along the channel dimension.  The region attention in RaGLU is also a kind of channel attention .All the operation about the spatial gathering remains conventional 3x3 convolution. So where the global receptive field exhibits?

**Questions:**

The questions have been listed in the Weaknesses.

---

> ### Author Response · Authors · 2024-11-25
> **Response to Reviewer pPeK**
>
> We express our sincere gratitude for your valuable time and constructive feedback!
>
> **[W1] Limited performance gains**
>
> Previous methods generally show limited improvements, **typically around 0.3%**. Our improvement is more significant than previous methods, as evidenced by Table 1 and Figure 1:
>
> + Table 1 shows that, in the Tiny and Small settings, RaCNN achieves significant performance improvement with fewer parameters and higher efficiency. For example, RaCNN-S **outperforms InceptionNeXt-S [1] by 0.4%** (83.9% vs 83.5%) while using only **half the FLOPs, 57% of the parameters**, and offering **a higher throughput**. Compared to MogaNet-S [2], which has similar FLOPs and parameter number, RaCNN-S provides a **0.5% improvement** (83.9% vs 83.4%) with **double the throughput**. Many other baselines show much smaller performance gains with lower efficiency.
> + Figure 1 illustrates the overall performance of state-of-the-art CNNs from recent years. It is evident that RaCNN achieves a more noticeable improvement across all variants compared to the baselines, pushing the Accuracy-FLOPs curve towards the upper-left corner by a greater margin.
>
> Besides, on lightweight architectures, RaCNN has greater performance gains than lightweight baselines, as demonstrated in Table 3. When FLOPs $\le$ 1G, RaCNN achieves a **1.6% accuracy improvement** over the state-of-the-art UniRepLKNet. In contrast, previous lightweight models show limited improvements, **typically below 1.0%**.
>
> Moreover, all reviewers unanimously agree that RaCNN demonstrates excellent performance across various visual tasks.
>
> **[W2] Insufficient ablation experiments, including the impact of cosine distance vs. inner-product in Dynamic PWConv**
>
> For RaCNN-T, replacing cosine distance with inner-product results in a decrease in accuracy from 82.9% to 82.7% on ImageNet.
>
> For additional ablation studies:
>
> + If you are interested in the impact of region size and region partitioning strategy, please refer to the response to `[W3,Q3]` of Reveiwer `zCMe`.
>
> **[W3] Doubt about global receptive field**
>
> + **RPWConv:** In RPWConv, Dynamic PWConv is employed in each sparse global window, and dynamic weights are generated based on global sparse spatial features that provide a global view. Therefore, the generated weight can capture information spread across the entire image.
> + **RaGLU:** RaGLU applies sparse global pooling to downsample spatial features. Each position in the resulting low-resolution features is averaged from scattered sparse positions in the input features, thus preserving global spatial information.
>
> In both modules, each spatial position in the output features is calculated based on the global sparse spatial features (i.e. the positions with the same color in Figure 4(c)), hence it possesses a global receptive field. Therefore, our proposed techniques are consistent with the paper's title.
>
> For more details, please refer to `[Common Question 1]`.
>
> ---
>
> **Reference**
>
> [1] InceptionNeXt: When Inception Meets ConvNeXt. CVPR 2024
> [2] MogaNet: Multi-order Gated Aggregation Network. ICLR 2024

---

### Author Response · Authors · 2024-11-25
**General response (part 1)**

We are grateful to all the reviewers for their efforts and valuable feedback!

We appreciate all the reviewers for their positive evaluation as below:

+ **Valuable motivation:** Our effort to overcome the high computational complexity of large kernel is valuable (Reviewer `PZ92`).
+ **Novel methods:** The proposed RPWConv and RaGLU incorporate a novel region-based dynamic weighting approach (Reviewer `zCME`), helping RaCNN to better capture global cues and local information (Reviewer `pPeK`, `EQAG`, `zCME`).
+ **Outstanding empirical performance:** Extensive experiments (Reviewer `CKsH`) effectively prove that RaCNN outperforms state-of-the-art CNNs and ViTs with lower computational complexity (Reviewer `pPeK`, `zCME`). The efficiency is an advantage of RaCNN (Reviewer `CKsH`).
+ **Versatility and scalability:** RaCNN achieves excellent performance across various tasks (Reviewer `pPeK`, `EQAG`, `CKsH`, `zCME`, `PZ92`) and different variants (Reviewer `zCME`), demonstrating robustness and effectiveness (Reviewer `pPeK`, `zCME`).

Here we answer some common questions shared by several reviewers.

**[Common Question 1] Some reviewers have questions about how RaCNN achieves global receptive fields with RPWConv and RaGLU**

Global receptive fields refer to the ability to focus on the entire image. In both modules, **sparse global view** plays a crucial role to obtain global receptive fields. Each spatial position in the output features is computed based on global sparse spatial features (i.e. positions with the same color in Figure 4(c)), enabling global receptive fields.

RPWConv: Given a feature map with shape $C\times H\times W$, RPWConv firstly divides it into $\frac{H}{rs}\times \frac{W}{rs}$ square spatial regions, where $rs$ is the region size. Then, pixels at the same relative position within all regions are sampled to form a **sparse global window** (renamed from "dilated window" in the original manuscript to avoid discrimination mentioned by Reviewer `EQAG`) and these pixels are aggregated into a vector in each channel. In other words, positions with the same color form a sparse global window. Dynamic PWConvs are applied in each sparse global window to generate dynamic weights tailored for that window and perform channel mixing. **The global perception capability of RPWConv stems from the sparse global window**, as each window gathers information from a global range and offers a global view of the image. Moreover, compared to Dynamic RWPConv, which shares the same dynamic weights across all positions, RPWConv generates unique weights for each sparse global window.

RaGLU: **Its ability to capture long-range dependencies derives from the sparse global pooling**. In the Region Attention, **sparse global average pooling** is employed to downsample spatial features. Each position in the resulting low-resolution features is averaged from scattered spatial positions in the input features, thus embedding global spatial cues.

To clarify this common question, we have provided additional explanations in the revised PDF. Figures 4 and 5 have been updated for better clarity, and we have added captions to Figure 5 to better explain the sparse global pooling in Region Attention.

---

> ### Author Response · Authors · 2024-11-25
> **General response (part 2)**
>
> **[Common Question 2] Several reviewers have questions about the visualization of RaCNN and other models in Figure 2, and doubt the conclusion that RaCNN can capture both global and local information**
>
> Some reviewers question the claim that other models "introduce excessive visual noises". A prior observation is that images in ImageNet often have semantic regions in the center (that's why center cropping is commonly used during testing). Consequently, **noise typically appears in edge and background areas**. Other models focus excessively on these noisy areas, so their receptive fields appear more "global". For instance, in Figure 2(a), SLaK [1] assigns excessive focus on edge areas, which is unreasonable. In Figure 2(c), InceptionNeXt [2] allocates excessive weight to strip-shaped regions due to its strip-shaped convolution. While UniRepLKNet [3] provides better global ERF in Figure 2(b), it still overemphasizes background areas. In comparison, our RaCNN focuses more on the central region while assigning smaller weights to edge and background areas, thereby avoiding excessive noise while maintaining its ability to capture global cues. Notice that the receptive weights are larger in the bottom-left corner, indicating that RaCNN is able to cover the entire image.
>
> Some reviewers think that the visualization is limited in a single layer. In fact, we utilize the visualization tool provided by RepLKNet [4]. Its illustration of ERF is based on the analysis of the **entire network** and its corresponding pre-trained weights rather than focusing on individual layers. Therefore, this method does not support single-layer analysis. To address this limitation, we will train multiple models with fewer layers to enable a more detailed analysis at the layer level in the future.
>
> Besides, from the perspective of implementation of large receptive fields, other methods use large kernels ($\ge$11), whereas our RaCNN does not. Instead, we only use standard $3\times 3$ convolutions combined with our proposed novel techniques to simulate large kernels, achieving a global receptive field. This design offers significant speed advantages, resolving the computational complexity of large kernels, as demonstrated in Table 1.
>
> We realize that our explanation for Figure 2 could be clearer, so we have added more explanations in the revised PDF.
>
> ---
>
> **Reference**
>
> [1] More ConvNets in the 2020s: Scaling up Kernels Beyond 51x51 using Sparsity. ICLR 2023
> [2] InceptionNeXt: When Inception Meets ConvNeXt. CVPR 2024
> [3] UniRepLKNet: A Universal Perception Large-Kernel ConvNet for Audio, Video, Point Cloud, Time-Series and Image Recognition. CVPR 2024
> [4] Scaling Up Your Kernels to 31×31: Revisiting Large Kernel Design in CNNs. CVPR 2022

---

### Meta-Review · Area_Chair_nY2Y · 2024-12-20

**Metareview:**

The paper proposes  a Region-aware CNN that achieves a global receptive field without requiring extra complexity. It shows good scalability and surpasses some of existing sota lightweight models. The reviewers raised concerns about limited technical contributions, incremental improvement, insufficient ablation experiments, mechanism to capture global information, confusion about convolution and local attention. Despite the authors' efforts in the rebuttal, some of the concerns remain unresolved. The final scores of the paper are unanimous negative. Therefore, the paper is rejected based on the reviewers' feedback.

**Additional Comments On Reviewer Discussion:**

The reviewers raised concerns about limited technical contributions, incremental improvement, insufficient ablation experiments, mechanism to capture global information, confusion about convolution and local attention. Despite the authors' efforts in the rebuttal, some of the concerns remain unresolved.

---

### Decision · Program_Chairs · 2025-01-22

Reject